# Loss of effort in chronic low-back pain patients: Motivational anhedonia in chronic pain

Samuel Alldritt[1], Mohammad Jammoul[2], Meena Makary[3,4], Susanne Becker[5], Daniel Maeng[2], Brian Keane[2], David Zald[6], Paul Geha[7,8]*

1 Child Mind Institute, New York, New York, United States of America, 2 Department of Psychiatry, School of Medicine and Dentistry, University of Rochester, 300 Crittenden Blvd. Rochester, New York, United States of America, 3 Systems and Biomedical Engineering Department, Faculty of Engineering, Cairo University, Cairo, Egypt, 4 Athinoula A. Martinos Center for Biomedical Imaging, Department of Radiology, Massachusetts General Hospital, Charlestown, Massachusetts, United States of America, 5 Clinical Psychology, Department of Experimental Psychology, Heinrich Heine University Düsseldorf, Düsseldorf, Germany, 6 Department of Psychiatry, Robert Wood Johnson Medical School, Rutgers University, 7 Department of Neuroscience, School of Medicine and Dentistry, University of Rochester, 8 Department of Neurology, School of Medicine and Dentistry, University of Rochester

* paul_geha@urmc.rochester.edu

## Abstract

The motivational and affective properties of chronic pain significantly impact patients' lives and response to treatment but remain poorly understood. Most available phenotyping tools of chronic pain affect rely on patients' self-report. Here we instead directly studied the willingness of chronic low-back pain (CLBP, n = 82) patients to expend effort to win monetary rewards available for wins at different probabilities and different levels of difficulties in comparison to matched pain free controls (n = 43). Consistent with the hypothesis of "negative hedonic shift" in chronic pain we observed that CLBP patients are significantly less willing than pain free controls to expend effort to go for high cost/high reward choices and their reported low-back pain intensity predicted increased effort discounting. Patients' performance was not explained by their self-reported depressive symptoms. Our results present new behavioral evidence characterizing the nature of anhedonia in chronic pain and highlight the importance of recognizing and assessing diminished motivation as an integral component of the chronic pain experience.

## Introduction

More than four decades ago Melzack and Casey emphasized that motivational and affective properties may be the "most important part" of the problem of pain [1]. Fast forward five decades into the future, we are starting to reconceptualize the whole chronic pain experience as consisting mainly of a negative affective and motivational experience [2–6]; in other words, chronic pain is not a continuum of acute pain and is characterized by distinct adaptations in the peripheral and central nervous system [7],

**Data availability statement:** All relevant data are within the paper and its Supporting information files.

**Funding:** The data we collected was supported by National Institute on Drug Abuse (NIDA) (5K08DA037525), National Institute of Neurological Disorders and Storke (NINDS) (R21NS1181162, R01NS127901), the Psychiatry Department at the Yale School of Medicine, and the Psychiatry Department at the University of Rochester Medical Center. The funders had no role in study design, data collection and analysis, decision to publish, or preparation of the manuscript.

**Competing interests:** The authors have declared that no competing interests exist.

which give rise to a negative affective condition more similar to depression, anxiety, or post-traumatic stress disorder than to an acute burn for example [8].

Chronic pain is characterized by a "negative hedonic shift" reflected in increased negative affect and decreased motivation to seek positive rewards [6]. This observation is in stark contrast with the finding that acute pain administered in pain free controls increased their motivation to seek positive rewards [9]. Nevertheless, the nature of the affective experience in chronic pain is still not well understood [10,11], and the behavioral approaches to measure chronic pain affect are still lacking. Questionnaires have been developed to assess the depressive and anxiety symptoms or other negative emotional states which co-occur with chronic pain [12–14]. These tools while useful in phenotyping patients [15], are mainly based on clinical observations and loosely linked to the neurobiology of chronic pain [1]. In addition, while it is important to assess co-morbid depression and anxiety, these symptoms are not necessarily equivalent to chronic pain affect. Importantly, most if not all these tools were in fact developed prior to the past two decades where the role of supraspinal neural circuitries in chronic pain came into scrutiny [2,5,16].

Pre-clinical and brain imaging studies of chronic pain have now demonstrated the role of the cortico-striatal circuitry in tracking clinical pain intensity and affect [17–25] and in predicting the transition from sub-acute to chronic pain [26–28]. Mesolimbic dopaminergic cells of the ventral-tegmental area projecting to the nucleus accumbens shell and core show region specific alterations in their firing patterns in animal models of chronic pain [29–32]. The cortico-striatal circuitry mediates reward processing and is implicated in both the subjective "liking" of rewards, and the willingness or motivation to seek rewards (reward "wanting") [33–36]. Reward processing has been hypothesized to be altered in chronic pain patients since Hippocrates [37], but studies directly addressing this hypothesis in humans remain limited. We and others have demonstrated that chronic pain is associated with subjective anhedonia [4,38–40]; in addition, we and others have observed disrupted decision making when chronic pain patients are offered choices of rewarding stimuli [38,39,41,42]. Here we specifically test the willingness of chronic pain patients to expend effort to obtain a monetary reward occurring at different probabilities using the Effort Expenditure for Rewards Task (EEfRT) [43], which was developed to objectively measure motivation in humans.

## Materials and methods

### Ethics statement

The study was approved by the Yale University and University of Rochester Institutional Review Boards and written informed consent was obtained from all participants.

### Data sources and participants

Data used in this work was collected at two different institutions: Yale University between 01/31/2018 and 11/15/2018, and University of Rochester Medical Center between 01/20/2021 and 03/05/2024 in Dr. P.G.'s laboratory. None of the behavioral

data presented in this work was published previously. Thirteen chronic low-back pain patients and twelve healthy controls from the Yale dataset were included in two prior, separate studies that focused on brain imaging [27,44]. Eighty-two patients with chronic low-back pain (CLBP) were recruited into the study if they had low-back pain below the 10th thoracic vertebra, for at least one year with a pain intensity ≥ 30/100 on a visual analogue scale (VAS) and no other chronic pain, neurologic, or psychiatric conditions. Therefore, patients were excluded if they reported current history of more than moderate depression, defined as a score > 19 on the Beck Depression Index [45], or history of traumatic brain injury, chronic psychiatric conditions, chronic inflammatory conditions (e.g., rheumatoid arthritis), or current ongoing chronic pain other than low-back pain. The same eligibility criteria were used to recruit 44 pain free controls who, in addition, denied any history of clinical pain. Participants completed the tasks between 9 and 11 am in the lab; after obtaining written consent a urine drug screen was obtained. Next, height and weight were directly measured in the lab using a Detecto (Inc.) scale.

## Demographic and clinical data

Participants completed questionnaires to assess handedness, depression, and anxiety. As shown in Table 1, the Beck Depression Inventory (BDI) [45] and Beck Anxiety Inventory (BAI) [46] were available for most participants, while PROMIS [47] Depression and Anxiety scale were available for the remaining ones. Participants also reported their pain experience using the McGill Pain Questionnaire (sf-MPQ) [48], and the Pain Catastrophizing Scale (PCS) [14].

## The Effort Expenditure for Rewards Task (EEfRT)

EEfRT has been thoroughly described by Treadway et al.[43]. Briefly, EEfRT is a multi-trial task where participants are given an opportunity on each trial to choose between two different task difficulty levels associated with varying levels of monetary reward. Effort expenditure on this task is inversely related to anhedonia [43] and depressed patients are less willing to expend effort than pain free controls on this task [49]. Each trial presents the participant with a choice between, a 'hard task' ((high cost/high reward (HC/HR) and an 'easy task' (low cost/low reward (LC/LR)) option, which require different amounts of speeded manual button pressing. For easy-task choices, subjects are eligible to win the same amount, $1.00, on each trial if they successfully complete the task. For hard-task choices, subjects are eligible to win higher

**Table 1. Demographic and clinical characteristics.**

|  | CLBP | HC |  | Missing |
|---|---|---|---|---|
|  | n = 82 | n = 43 | P-Value |  |
| Females | 19 | 23 | 0.10◇ | 0/0 |
| Age (yrs.) | 42.9 ± 3.0 | 39.2.9 ± 2.6 | 0.24 | 0/0 |
| BMI (Kg/m²) | 26.5 ± 0.8 | 26.5 ± 0.9 | 0.96 | 0/0 |
| Handedness | 71R/7L/2A | 35R/6L/1A | 0.642◇ | 2/1 |
| Education (yrs.) | 16.1 ± 0.4 | 17.3 ± 0.4 | 0.04 | 0/0 |
| Pain duration (yrs.) | 8.0 ± 1.3 | – |  | 0/0 |
| Employed | 66.1% | 65.2% | 1.0◇ | 20/20 |
| VAS | 47.7 ± 2.6 | – |  | 0/0 |
| MPQ | 10.6 ± 7.2 | – |  | 11/0 |
| PCS | 13.8 ± 10.5 | – |  | 21/0 |
| BDI | 7.8 ± 1.0 | 2.1 ± 1.1 | $< 10^{-3}$ | 27/6 |
| BAI | 6.2 ± 0.8 | 2.6 ± 0.9 | < 0.01 | 27/6 |
| PROMIS Anxiety | 14.5 ± 5.3 | 11 ± 2.5 | 0.14 | 56/37 |
| PROMIS Depression | 12.6 ± 5.4 | 9 ± 1.6 | 0.13 | 56/37 |

◇, Chi-square test

amounts that vary randomly from trial to trial within a range of $1.24 – $4.30 ("reward magnitude"). The win during any task is not however guaranteed but subjects are given accurate probability cues at the beginning of each trial with high (88%), medium (50%) and low (12%) probability of win.

## Statistical analysis

*EEfRT Data Reduction.* Because subjects could only play for 20 minutes, the number of trials completed during that time varied from subjects to subject. Time limitation of the EEfRT serves to avoid severe subject fatigue. For consistency only the first 50 trials were used. Nevertheless, there was no group difference in the number of trials between pain free controls (mean ± SEM = 67.1 ±) and CLBP patients (70.1 ± 1.6) (p = 0.20, t-score (degrees of freedom) $t_{(76)}$ = − 1.29 unpaired t-test).
*Analysis Method 1.* The EEfRT data was analyzed following two different approaches. In the first approach, the mean proportion of the hard task choices (HC/HR) was calculated at each probability and compared across levels of probability (i.e., 12%, 50%, and 88%), or calculated at each reward magnitude and compared across reward magnitudes (i.e., < $2.5, $2.5-$3.5, and > $3.5) as within subject factors, and between groups (i.e., CLBP vs pain free controls) using generalized least squares (GLS) model with heteroskedastic error terms and autoregression 1 serial correlation [50]. Age, sex, site, and years of education were included in the model as confounders.
*Analysis Method 2.* In the second approach we applied a computational model described by Cooper et al. [51] to analyze effort based decision making. Each trial of the EEfRT provides subjects with two pieces of information to consider when selecting between the high and low effort options: the reward magnitude for the high effort option and the probability of winning. To estimate the effort discounting rate (k) we fit the "full subjective value (SV) model" in which SV = RP$^h$- kE, where R ($1–4.30) is the reward magnitude, P is the probability of winning, E is the amount of effort (0.3 for low effort and 1.0 for high effort), and h is the extent to which the subjects weigh subjective value based on probability [51]. In this model, both *h* and *k* are free parameters. Effort perceived as extremely costly is reflected in a higher value of *k*; weighting for probability is captured by *h*. This model assumes that subjects consistently incorporate both trial-wise reward and probability when selecting. The SVs were translated into probabilities of selecting each option using the Softmax decision rule equation [52] implemented in MATLAB,

$$p(hard) = \frac{e^{SVhard.t}}{e^{SVhard.t} + e^{SVeasy.t}}$$

Where *t* is an inverse temperature parameter that reflects a tendency to favor options with high SVs. To test for group differences between CLBP patients and pain free controls, we compared *k* (effort discounting), *h* (subjective value), and t (inverse temperature) parameters between groups using unpaired t-test corrected for age and sex.

## Results

### Sample characteristics

CLBP patients had an average pain duration of 8.0 ± 1.3 years (mean ± SEM) and reported an average low-back pain intensity of 42.8 ± 2.2 on the VAS. CLBP patients and pain free control subjects did not differ in age, sex distribution, or body mass index (BMI) (Table 1). CLBP patients reported on average two years of education less than the pain free controls and that difference was significant (CLBP patients, 15.8 ± 0.3 years of education; pain free controls, 17.0 ± 0.4 years of education, t-score (degrees of freedom) $t_{(122)}$ = 2.78, unpaired t-test, p = 0.006). CLBP patients' BDI and BAI scores were significantly larger than those of pain free control subjects (BDI: CLBP patients, 7.2 ± 1.1; pain free control, 2.1 ± 0.5, $t_{(90)}$ = − 3.7, p < 10$^{-3}$; BAI: CLBP patients, 6.8 ± 1.1; pain-free controls, 2.4 ± 0.6, $t_{(93)}$ = − 3.21, p = 0.002). A sub-group of participants (27 CLBP patients and 6 pain-free controls) did not have BDI or BAI because they were part of a third study but reported mood and anxiety symptoms on the Hospital Anxiety and Depression Scale (HADS). By design, patients on

opioids were not included in this study. In addition, the only allowed psychoactive medications were antidepressants and gabapentinoids. We present the distribution of patients by medication in S1 Fig.

## Results of group comparisons across probability and reward levels in the EEfRT

We tested two GLS models corrected for age, sex, sites, and years of education for the main effect of group (CLBP vs. pain free controls), and for the interaction between group and levels of probability (model 1) and group and magnitude of reward respectively (model 2) on the preference for the high cost/high reward (HC/HR) options. Using model 1 we found a significant effect of group on the preference of the HC/HR option ($t_{(364)}$ = 2.61; p = 0.0094), and a significant interaction ($t_{(364)}$ = − 4.51, p < $10^{-4}$) between group and probability levels stemming from the observation that CLBP patients expended significantly less effort than controls for options with higher probabilities of win (i.e., at 50% and 88% probability) (Fig 1A, S2 FigA, S1 Table). Using model 2, we found a significant effect of group on the preference of the HC/HR option ($t_{(364)}$ = − 2.34, p = 0.02), and a significant group by reward magnitude interaction ($t_{(364)}$ = 2.89, p = 0.0041) (Fig 1B, S2 FigB, S1 Table). Examining the graph in Fig 1B we observe that CLBP patients expended less effort than pain free control subjects especially when faced with low reward magnitude but that they recovered faster as the reward magnitude increased. Some of our CLBP patients (n = 13) were prescribed psychoactive medications (i.e., antidepressants and/or gabapentinoids, S1 Fig.) which may affect their performance on the EEfRT. Therefore, we repeated our analysis after excluding these patients. The results remain unchanged (S2 Table)

Because some participants had missing BMI and BDI was not collected on all participants, we repeated the GLS analysis after adding BMI and BDI in separate analyses as variables of no interest. Adding BMI does not change the results described above. Only group effects in model 1 change after adding BMI or BDI to model 1, which examines group effect across the 3 levels of probability of wins; it was no longer statistically significant after adding BMI (p = 0.052) or after adding BDI (p = 0.61); nevertheless, the group x probability interaction remained significant (p < 0.01) showing that CLBP patients were less willing to expend effort as the probability of wins increased. These analyses are presented in detail in S2 Table with the **Supporting Information**.

## Results obtained from fitting the "full subjective value model"

The "full subjective value model" [51] was applied to obtain the parameters $k$ (effort discounting), $h$ (subjective value), and $t$ (inverse temperature). None of these parameters showed significant differences between CLBP patients and pain free controls when compared after correcting for age, sex, sites, and years of education (mean $k$ ± SEM in CLBP = 3.30 ± 0.35; pain free controls = 2.57 ± 0.44; p = 0.38 tested against 5000 permutations; mean $h$ ± SEM in CLBP = 11.9 ± 2.4, pain free controls = 13.4 ± 4.4, p = 0.92 (non-parametric permutations); mean $t$ ± SEM in CLBP = 3.4 ± 0.4 and in pain free

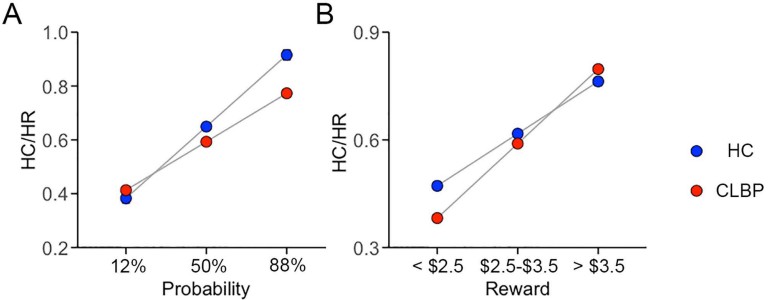

**Fig 1. Plots showing the adjusted average ± SEM proportion of HC/HR choices (y-axis) on the first 50 trials of the EEfRT as a function of probability levels (A) and reward magnitudes (B) (x-axis) for CLBP (red) and pain free controls (blue) subjects.**

controls = 2.4 ± 0.49, p = 0.46 (non-parametric permutations)). However, CLBP intensity reported on the visual analogue was positively correlated to the effort discounting parameter $k$ (spearman-$\rho$ = 0.24, p = 0.03) (Fig 2) suggesting therefore that the more the pain the the less the patients were willing to expend effort.

## Discussion

In this study we present evidence that CLBP patients exhibit behavior characteristic of motivational anhedonia, as measured by an objective cost/benefit decision-making task [49]. Patients suffering from CLBP are less willing to expend effort to obtain monetary rewards, even with increasing magnitude or increasing probability of wins. Furthermore, reported CLBP intensity was directly related to the effort discounting rate, supporting the observation that clinical pain hinders patients' motivation to seek rewards. These observations are consistent with pre-clinical [5,22,24,25,53] and brain imaging findings showing that chronic pain patients' motivational pathways are disrupted [2]. They are also in line with our previous findings of perceptual anhedonia in CLBP patients when presented with highly palatable foods [38,39], and reports of anhedonia [4] and disrupted emotional decision making from the literature [41,42]. This motivational anhedonia is not explained by patients' depressive symptoms as reported on the Beck's Depression Inventory. Recruitment of CLBP patients with no significant clinical depression by design may have, in fact, underestimated the loss of motivation in chronic pain, and may also explain the absence of significant differences in the parameters of the SV model [51]. The selection of CLBP patients with minimal or no depression and anxiety symptoms was, however, necessary, to avoid confounds associated with the psychopathologies.

Negative affect is a major symptom of chronic pain and is often a significant negative predictor of the resolution of pain or of analgesic success [54]. Nevertheless, how the affective experience of chronic pain patients differs from that of other conditions such as depressive or anxiety disorders remains unclear. Our current and previous [38,39] results show that the loss of perceptual pleasure in experiencing or seeking rewards characterizes patients with chronic pain even in the absence of clinically significant symptoms from major affective or anxiety disorders. Consistently, Garland et al. [4], observed that anhedonia measured using Snaith-Hamilton Pleasure Scale [55] in chronic pain patients with co-morbid depression cannot be explained only by the latter diathesis. Anhedonia is conceptualized not only as a "marked and consistent decrease in interest or pleasure in almost all daily activity", but also as a loss of interest to act to seek such pleasure, and is called amotivation [56,57]. CLBP patients exhibit therefore both aspects of anhedonia independently of

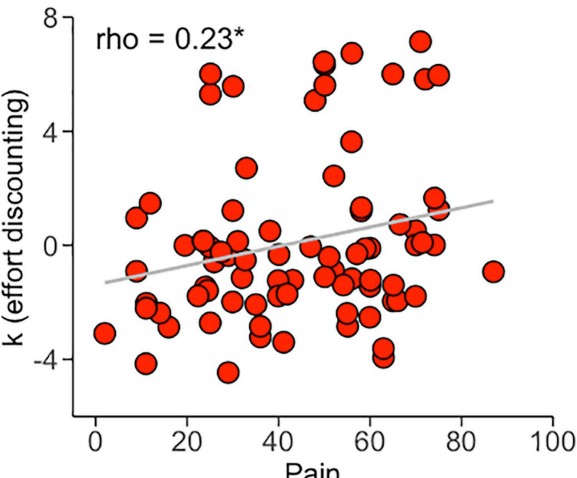

**Fig 2. Regression plot showing how effort discounting $k$ increases linearly (Spearman *rho*) with reported CLBP intensity.** *, p < 0.05.

reported symptoms of depression and anxiety. This in turn suggests that the negative affective symptoms associated with chronic pain may be the phenotypic expression of a specific disruption of cortico-striatal circuitry, distinct from that seen in affective disorders [8]. In contrast to chronic pain, acute painful stimulation delivered to pain free healthy participants increases motivation to seek monetary rewards and does not change hedonic reactions to these rewards [9]. Thus, the loss of motivation observed in chronic pain patients is unlikely to result solely from interference by negative sensory input.

Our results, considering that EEfRT is directly modulated by dopaminergic tone [58], suggest that striatal dopaminergic transmission may also be disrupted in CLBP patients. Preclinical data clearly indicate disruptions in mesolimbic [25,29–32] and nigro-striatal [53,59] dopaminergic transmission in chronic pain. However, evidence in chronic pain patients is limited [60–62]. A few positron emission tomography studies have reported decreased dopamine binding potential in the striatum of chronic pain patients compared to pain free controls [63], including one study on CLBP patients [60]. Indirect evidence from brain imaging studies also supports this hypothesis, as CLBP patients consistently show alterations in ventral striatal activity [27]. Additionally, the connectivity of this region to the ventro-medial prefrontal cortex tracks back-pain intensity [17] and predicts the transition from sub-acute to chronic pain [26,28]. We have previously observed a strong and positive relationship between hedonic measures and ventral striatal (accumbens) volume in CLBP patients, but not in pain free controls, when these subjects reported their liking of highly palatable foods [39]. Thus, the hedonic experience of chronic pain patients becomes closely associated with the properties of the mesolimbic system, which is increasingly involved in the patients' pain experience. This finding aligns with the established role of the accumbens and ventro-medial prefrontal cortex in cost/benefit processing [64,65] as they integrate internal state -in this case, back pain intensity- with the cost of the effort to guide decision-making. However, a direct link between these neuroimaging findings and disrupted dopaminergic transmission has yet to be established. The observation that modulation of dopaminergic tone using dopaminergic agonists can significantly reduce pain in fibromyalgia patients more than placebo [66] and can prevent the transition to CLBP in female sub-acute low-back pain patients [67] suggest that the observed neuroimaging findings may indeed reflect a disruption in striatal dopaminergic transmission.

In addition to the dopaminergic disruption that may mediate the observed motivational deficit in CLBP patients, a disruption in opioid transmission is another potential neurochemical pathway that can explain this anhedonia. Patients with chronic pain have shown decreased binding potential of opioid ligands in the ventral striatum [68–70]. Opioid receptors, which are plentiful in this sub-cortical area, [71,72], play a significant role in pain control, hedonic processing, and negative affect as established in various studies [23,33,73,74]. Opioid binding in the ventral striatum is thought to contribute to hedonic encoding of rewards [75], and to the behavioral responding to reward predictive cues [76]. Consistent with this role and the plasticity observed in the striatum of chronic pain patients [5,6,77–79] our current and previous [38,39] work show that CLBP patients exhibit disruptions in hedonic processing whether they are asked to report their subjective ratings of pleasure or to work for rewarding stimuli like money.

Motivational and hedonic deficits are of high clinical significance because, in addition to the loss of well-being, they may lead to the emergence of other co-morbidities often observed to be associated with chronic pain such as substance use disorder [80–82], obesity [83], and depression [84]. Anhedonia is associated with substance use disorders and is a target symptom for relapse prevention [3,85,86]. Interestingly, reports of subjective anhedonia are increased only in chronic pain patients misusing opiates but not in patients taking these medications as prescribed, suggesting that anhedonia predates substance misuse [4]. Anhedonia is also associated with weight gain [87,88], and evidence suggest that this may be a mechanism underlying the increased prevalence of obesity in chronic pain patients. Consistent with this hypothesis, hedonic ratings predict caloric intake in hungry and satiated pain free controls; however, this relationship is disrupted in chronic pain patients [38,39].

The data collection for this study was completed by different experimenters at two different locations. Therefore, despite accounting for potential confounders that we expect to affect the results such as age, sex, years of education, depression scores, and adding a dummy variable for site, it is possible that other sources of variability were not accounted for. Further studies are needed to corroborate our findings.

In conclusion, our results provide behavioral evidence that chronic pain patients exhibit motivational deficits similar to those observed in individuals with major depressive disorder [49]. However, this loss of motivation cannot be explained by self-reported depressive symptoms in the chronic pain patients. These findings highlight the importance of recognizing and assessing diminished motivation as an integral component of the chronic pain experience.

## Supporting information

**S1 Fig. Bar-plot illustrating the number of participants from each group taking medications.** Analgesics include acetaminophen, non-steroidal anti-inflammatory drugs, triptans, and gabapentinoids. Other Medications include all non-analgesic medications.
(DOCX)

**S2 Fig. Illustration of raw proportions of HC/HR choices for pain free healthy controls (HC) and chronic low-back pain (CLBP) patients using mean± SEM (left) and violin plots (right) while splitting the choices by probability of win (A) or reward magnitude (B).**
(DOCX)

**S1 Table. Adjusted mean proportion of the HC/HR choices for models 1 and 2.**
(DOCX)

**S2 Table. GLS analysis of high cost/high reward choices after accounting for BMI and BDI respectively.**
(DOCX)

**S1 Data. Raw data.**
(XLSX)

## Author contributions

**Conceptualization:** David Zald, Paul Geha.

**Data curation:** Samuel Alldritt, Mohammad Jammoul, David Zald, Paul Geha.

**Formal analysis:** Mohammad Jammoul, Paul Geha.

**Funding acquisition:** Paul Geha.

**Investigation:** David Zald, Paul Geha.

**Methodology:** Samuel Alldritt, Meena Makary, Daniel Maeng, David Zald, Paul Geha.

**Project administration:** Meena Makary.

**Writing – original draft:** Samuel Alldritt, Mohammad Jammoul, Susanne Becker, Daniel Maeng, Brian Keane, David Zald, Paul Geha.

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
