## [Decision Letter · Decision Letter 0]

6 May 2025

PONE-D-25-00836Loss of Effort in Chronic Low-Back Pain Patients: Motivational Anhedonia in Chronic Pain

PLOS ONE

Dear Dr. Geha,

Thank you for submitting your manuscript to PLOS ONE. After careful consideration, we feel that it has merit but does not fully meet PLOS ONE’s publication criteria as it currently stands. Therefore, we invite you to submit a revised version of the manuscript that addresses the points raised during the review process.

We look forward to receiving your revised manuscript.

Kind regards,

Roya Khanmohammadi, Ph.D

Academic Editor

PLOS ONE

Journal Requirements:

“Funding was provided to Dr. Paul Geha through this funders:  NIDA (5K08DA037525), NINDS (R21NS1181162, R01NS127901), the Psychiatry Department at the Yale School of Medicine, and the Psychiatry Department at the University of Rochester Medical Center.”

4. Please expand the acronym “NIDA, NINDS” (as indicated in your financial disclosure) so that it states the name of your funders in full.

5. Please note that funding information should not appear in the Acknowledgments section or other areas of your manuscript. We will only publish funding information present in the Funding Statement section of the online submission form. Please remove any funding-related text from the manuscript. 

**Additional Editor Comments:**

The reviewers found the study interesting and of potential value. However, substantial revisions are needed before a final decision can be made.

**Major Points Needing Revision:**

Clarify and illustrate fMRI analysis (masks, ROIs, whole-brain connectivity).Add the missing Table 1 and clarify any prior use/publication of data.Specify preregistration status and whether connectivity analyses were preplanned.Discuss limitations more thoroughly, including data collection across sites/times, use of NeuroCombat, resting-state scan timing, and motion artifacts.Report participants' medications and consider their impact.Revise the conclusion to avoid overstating causality.

**Minor Points:**

Clarify sample sizes in the abstract and methods.Improve clarity in the dataset descriptions and clinical scale results (suggested as a table).Adjust wording (e.g., replace “likely” with “may be”) where claims are too strong.Include participant employment status if available.Correct figure legend inconsistencies.

Reviewers' comments:

Reviewer's Responses to Questions

**Comments to the Author**

1. Is the manuscript technically sound, and do the data support the conclusions?

Reviewer #1: Partly

2. Has the statistical analysis been performed appropriately and rigorously? 

Reviewer #1: I Don't Know

3. Have the authors made all data underlying the findings in their manuscript fully available?

Reviewer #1: Yes

4. Is the manuscript presented in an intelligible fashion and written in standard English?

Reviewer #1: Yes

5. Review Comments to the Author

Reviewer #1: This is an interesting paper describing pooled data from participants with chronic low back pain (CLBP) and healthy controls who performed a reward effort assessment task (EEfRT) and a subgroup of these participants with resting-state fMRI data. The manuscript is mostly clear and of general interest. Some clarification of the methods and refinement of the ideas is needed to make sure that the message conveyed to the readers are translated appopriately and not over-stating what the results may suggest.

Major/moderate:

1. The description of the fMRI analysis needs some clarification. The masks for vmPFC and ACC from Neurosynth should be shown in the supplementary materials (or manuscript) because it is unclear how large these mask areas span within the brain, and only very focal spherical ROIs are shown in the Figure. On p. 10 lines 210-211, “to” is used twice in this sentence but I’m assuming that directionality of the connections is not being measured because the analysis is only a correlation between activity within the regions. On p. 10, lines 212 – 214, it is unclear from this description (but looks like it is the case in the figure) that the functional connectivity maps are from the selected ROIs to the whole brain, please specify, “whole brain” in this description somewhere.

2. I don’t see a Table 1 anywhere in my document as referenced in 3. Results.

3. Provide some information on whether (and if so where and how) any of the data from these 3 studies were published previously. Also, provide information regarding whether any of the participants (and which ones) have data published in other papers. Since it might be the case because the data are being combined from 3 studies, please specify, whether any of these data are used in this study as a secondary analysis.

4. Was there a preregistration plan for these analyses? Without this, it is somewhat challenging to know whether the behavioral data were previously analyzed in different ways (due to many possibilities of different conditions in the task, and possible prior analysis of smaller individual data sets). Additionally, were the 4 connectivity analyses planned a priori, or selected from other analyses? Please provide the details of these analyses, and whether any others were conducted on the fMRI data prior to the ones presented in the manuscript. P values should be provided for each connectivity result in the Supplement table (in addition to the ** p<0.01).

5. Some limitations should be pointed out in the manuscript. While age, site, and education were corrected for in the analysis, it doesn’t mean that everything is perfectly accounted for by this, so some description that the data were acquired over multiple times, locations, and by different experimenter is needed in the discussion. Also, it should be clarified in the discussion that the fMRI data were acquired during the resting-state, and not while performing the EEfRT task. Since resting-state fMRI activity can be influenced by prior tasks, it should also be described (maybe I missed it) and possibly evaluated/controlled for how the timing of fMRI data acquisition related to the timing of EEfRT task participation (i.e., were the scans all collected after the task session, or were any of the fMRI scans collected on a separate day?)

6. The authors used NeuroCombat to correct for site differences, however, more description would be helpful particularly because the citation [57]’s title indicates that the tool is for DTI data rather than fMRI data as used in this study.

7. In the concluding paragraph, the points jump around and are difficult to follow and are a bit too broad. Particularly, it is unclear what the second sentence means, so more details should be provided to provide a clearer vision of what the authors are suggesting, and how the data support this somewhat broad claim. The last line’s statement also infers a bit too much causality to the brain circuit changes being responsible for the negative affective experience. Please tone this down a bit.

8. Since the brain differences are fairly widespread, I have concerns that there could be motion contributing to these group differences. Motion averages should be included somewhere in the manuscript and analyzed to show that scan motion differences are not contributing to group differences observed.

9. I don’t see any descriptions of medications in the manuscript but assume that at least some of the patients were taking different medications. Please provide these data either descriptively or in a table. Additionally, medications should be at least somehow taken into account in the task and fMRI analyses (or listed explicitly as a limitation in the discussion), particularly if any of the participants were taking medications that can influence mood or affective symptoms.

Minor:

1. The abstract should specify the number of participants in the task analysis (full N), and that the functional connectivity analyses were conducted in a subgroup (specify N) of patients with fMRI data. Also, resting-state (i.e., not task-based) data should be clarified by adding “resting-state” before “functional connectivity”.

2. On p. 6 at the bottom of the page onto p. 7, the authors describe a third data set, however, as a reader I had to hunt through the third data set’s description to see what the unique aspects were compared to the second data set. If these sections can be re-arranged to state the third data set was the same as the second data set except for a few things, it would be much clearer.

3. It would be easier for the reader to digest the means and results for the BDI, BAI group comparisons in a Table. Please convert from the text to a table. HADS was collected from participants in the third data set; these data should be provided as well.

4. On P. 14 line 317, “likely” should be changed to “may be” or “might be”. It can’t really be concluded that the circuits are driving the affective symptoms; only that they are occurring together. Further, the patients had greater anxiety and depression scores than healthy controls in this study, so even though psychiatric conditions were exclusionary, the patients still showed greater levels of affective symptoms. The idea in line 317 is also somewhat problematic and unfounded because the patients did demonstrate worse affective symptoms than controls. This idea either needs to be more specifically linked to the data provided here and clarified, or changed to be more general and not make as specific of a claim.

5. Was employment/disability status collected? If so, these data should be provided along with the education levels. Employment could be of interest to align with the behavioral results.

6. Fig. 1 C legend: Lines 392-393 are not actually shown in the figure C. Need to either show the non-significant relationship in C, or adjust this statement in the legend.

6. PLOS authors have the option to publish the peer review history of their article (what does this mean? ). If published, this will include your full peer review and any attached files.

**Do you want your identity to be public for this peer review?** For information about this choice, including consent withdrawal, please see our Privacy Policy .

Reviewer #1: No

---

## [Author Response · Author response to Decision Letter 1]

3 Jul 2025

We thank the editor and reviewer for acknowledging the merit of our work and for their constructive feedback, which has significantly improved our manuscript. In response to the reviewer’s comments, we re-examined the medication lists of our participants and identified two patients who should have been excluded during screening. One patient was taking amphetamines, and another had a diagnosis of multiple sclerosis and was being treated with interferon beta. In addition, we discovered that a pain-free control participant had a previously undetected diagnosis of psoriasis. After excluding these individuals, our behavioral results on the EEfRT task remained unchanged. However, the correlation between the proportion of high-cost/high-reward choices and functional brain connectivity weakened. Specifically, the data previously presented in Figure 3 show now a correlation of ρ = – 0.24 (p = 0.18) for healthy controls and ρ = +0.24 (p = 0.04) for patients. This weaker effect and the higher p-value made us less confident in the robustness of the connectivity findings, and we have therefore chosen to remove these results from the manuscript. We have modified the results and discussion section accordingly. We believe this does not diminish the merit of the manuscript, as its primary contribution lies in the observed behavioral differences on the EEfRT.

Please find below our responses point by point.

Reviewer #1: This is an interesting paper describing pooled data from participants with chronic low back pain (CLBP) and healthy controls who performed a reward effort assessment task (EEfRT) and a subgroup of these participants with resting-state fMRI data. The manuscript is mostly clear and of general interest. Some clarification of the methods and refinement of the ideas is needed to make sure that the message conveyed to the readers are translated appropriately and not over-stating what the results may suggest.

We thank the reviewer for their appreciation of our manuscript.

Major/moderate:

1. The description of the fMRI analysis needs some clarification. The masks for vmPFC and ACC from Neurosynth should be shown in the supplementary materials (or manuscript) because it is unclear how large these mask areas span within the brain, and only very focal spherical ROIs are shown in the Figure. On p. 10 lines 210-211, “to” is used twice in this sentence but I’m assuming that directionality of the connections is not being measured because the analysis is only a correlation between activity within the regions. On p. 10, lines 212 – 214, it is unclear from this description (but looks like it is the case in the figure) that the functional connectivity maps are from the selected ROIs to the whole brain, please specify, “whole brain” in this description somewhere.

As mentioned above we have now removed the fMRI data from the manuscript.

2. I don’t see a Table 1 anywhere in my document as referenced in 3. Results.

We apologize we did not upload the table upon submission. We have now uploaded it.

3. Provide some information on whether (and if so where and how) any of the data from these 3 studies were published previously. Also, provide information regarding whether any of the participants (and which ones) have data published in other papers. Since it might be the case because the data are being combined from 3 studies, please specify, whether any of these data are used in this study as a secondary analysis.

No behavioral data was published previously from any of the 3 studies. 13 CLBP patients and 12 HC collected at Yale were part or previous papers examining brain biomarkers of chronic pain [2; 3] to test a completely different hypothesis. The analysis in this manuscript is not a secondary analysis. We have been collecting resting state fMRI and EEfRT data on all our chronic pain patients. The data is part of different studies, but it aims to answer the question raised in this manuscript—that of motivational anhedonia. We have now added this information in the methods’ section under “Data sources and participants”; we write:

“None of the behavioral data presented in this work was published previously. Thirteen chronic low-back pain patients and twelve healthy controls from the Yale dataset were included in two prior, separate studies that focused on brain imaging [2; 3].”

4. Was there a preregistration plan for these analyses? Without this, it is somewhat challenging to know whether the behavioral data were previously analyzed in different ways (due to many possibilities of different conditions in the task, and possible prior analysis of smaller individual data sets).

Additionally, were the 4 connectivity analyses planned a priori, or selected from other analyses? Please provide the details of these analyses, and whether any others were conducted on the fMRI data prior to the ones presented in the manuscript. P values should be provided for each connectivity result in the Supplement table (in addition to the ** p<0.01).

The study was not pre-registered. However, all possible analytic approaches to the EEfRT task were included in this manuscript, following both the original implementation of the task[5] and its subsequent application in other conditions involving anhedonia components[1]. Accordingly, we report two complementary approaches: Method 1, which examines group differences in the average selection of high-cost/high-reward trials, and Method 2, which compares group differences in the parameters of the subjective value model. Interim analysis of the data was performed with smaller number of subjects and yielded similar results to the ones presented in the manuscript.

Since the fMRI data was now removed, we will not need to address the connectivity choices.

5. Some limitations should be pointed out in the manuscript. While age, site, and education were corrected for in the analysis, it doesn’t mean that everything is perfectly accounted for by this, so some description that the data were acquired over multiple times, locations, and by different experimenter is needed in the discussion. Also, it should be clarified in the discussion that the fMRI data were acquired during the resting-state, and not while performing the EEfRT task. Since resting-state fMRI activity can be influenced by prior tasks, it should also be described (maybe I missed it) and possibly evaluated/controlled for how the timing of fMRI data acquisition related to the timing of EEfRT task participation (i.e., were the scans all collected after the task session, or were any of the fMRI scans collected on a separate day?)

The reviewer raises a very valid point, and we agree on some of these limitations. The reviewer is correct in that we cannot account for all possible confounders. However, we have tried to account for all the confounders that we expect to influence EEfRT such as age, sex, education, site, and depression scores. The data collection at multiple locations might be both a confounder but also a strength because our results are presented after adding a dummy variable for the sites. Additionally, since the data was collected by different people at different sites, the results are therefore unlikely to be researcher dependent. We added a statement in the discussion acknowledging this limitation at the end of the discussion we now write:

“The data collection for this study was completed by different experimenters at two different locations. Therefore, despite accounting for potential confounders that we expect to affect the results such as age, sex, years of education, depression scores, and adding a dummy variable for site, it is possible that other sources of variability were not accounted for. Further studies are needed to corroborate our findings”.

The lag between the fMRI acquisition and EEfRT data is no longer relevant with the revised version of the manuscript.

6. The authors used NeuroCombat to correct for site differences, however, more description would be helpful particularly because the citation [57]’s title indicates that the tool is for DTI data rather than fMRI data as used in this study.

The question about harmonization is no longer relevant because we removed the fMRI results.

7. In the concluding paragraph, the points jump around and are difficult to follow and are a bit too broad. Particularly, it is unclear what the second sentence means, so more details should be provided to provide a clearer vision of what the authors are suggesting, and how the data support this somewhat broad claim. The last line’s statement also infers a bit too much causality to the brain circuit changes being responsible for the negative affective experience. Please tone this down a bit.

Per the reviewer’s suggestion we have now improved the focus of the conclusion by concentrating on the diminished motivation in chronic pain patients; we have also clarified the second sentence; we now write:

“In conclusion, our results provide behavioral evidence that chronic pain patients exhibit motivational deficits similar to those observed in individuals with major depressive disorder [4]. However, this loss of motivation cannot be explained by self-reported depressive symptoms in the chronic pain patients. These findings highlight the importance of recognizing and assessing diminished motivation as an integral component of the chronic pain experience”.

8. Since the brain differences are fairly widespread, I have concerns that there could be motion contributing to these group differences. Motion averages should be included somewhere in the manuscript and analyzed to show that scan motion differences are not contributing to group differences observed.

The imaging data was removed from the manuscript.

9. I don’t see any descriptions of medications in the manuscript but assume that at least some of the patients were taking different medications. Please provide these data either descriptively or in a table. Additionally, medications should be at least somehow taken into account in the task and fMRI analyses (or listed explicitly as a limitation in the discussion), particularly if any of the participants were taking medications that can influence mood or affective symptoms.

The reviewer raises a very important point about medication intake which when re-examined helped us identify 3 participants that should have been excluded because they were taking medications that could directly affect their performance on EEfRT (1 patient on amphetamine, 1 patient on interferon beta) or because of a previously missed exclusionary criterion (one pain free control with psoriasis). We now present all the medications that patients have been taking in S1Fig. Please note that patients on opioids were excluded by design. To test whether medication intake affect the EEfRT differences between CLBP and pain free controls we repeated the analysis presented in the Figure 1 after removing 13 CLBP patients who were taking psychoactive medications (i.e., antidepressants and/or gabapentinoids). The results remain unchanged and are now presented in S2Table.

Minor:

1. The abstract should specify the number of participants in the task analysis (full N), and that the functional connectivity analyses were conducted in a subgroup (specify N) of patients with fMRI data. Also, resting-state (i.e., not task-based) data should be clarified by adding “resting-state” before “functional connectivity”.

We have now added the number of participants in the abstract.

2. On p. 6 at the bottom of the page onto p. 7, the authors describe a third data set, however, as a reader I had to hunt through the third data set’s description to see what the unique aspects were compared to the second data set. If these sections can be re-arranged to state the third data set was the same as the second data set except for a few things, it would be much clearer.

Because the imaging data has been removed, the section that the reviewer is referring to was also removed. As noted under “Data Sources and Participants” , we describe the data source and specify the dates and sites of data collection.

3. It would be easier for the reader to digest the means and results for the BDI, BAI group comparisons in a Table. Please convert from the text to a table. HADS was collected from participants in the third data set; these data should be provided as well.

We have now included Table 1, which was omitted in the previous submission. We included PROMIS anxiety and depression for the third data set. Furthermore, we have reported the number of participants who were missing PROMIS or BDI/BAI in the table. We corrected our earlier version; we had stated we collected HADS when in fact we collected PROMIS.

4. On P. 14 line 317, “likely” should be changed to “may be” or “might be”. It can’t really be concluded that the circuits are driving the affective symptoms; only that they are occurring together. Further, the patients had greater anxiety and depression scores than healthy controls in this study, so even though psychiatric conditions were exclusionary, the patients still showed greater levels of affective symptoms. The idea in line 317 is also somewhat problematic and unfounded because the patients did demonstrate worse affective symptoms than controls. This idea either needs to be more specifically linked to the data provided here and clarified or changed to be more general and not make as specific of a claim.

The reviewer is correct in pointing that the conclusion that the circuits are driving the affective symptoms does not have strong evidence but may be suggested by the results. We therefore changed “likely” to “may be” per the reviewer’s suggestion. Nevertheless, the idea that patient’s increased level of anxiety and/or depressive symptoms do not explain our findings are supported by our approach and our results. First, as the reviewer mentions, we excluded patients suffering from more than mild symptoms of depression and/or anxiety. Second, when we corrected for depression scores, we obtained a very similar results despite having significantly smaller sample size.

5. Was employment/disability status collected? If so, these data should be provided along with the education levels. Employment could be of interest to align with the behavioral results.

We have collected employment status. By design, we exclude patients on disability. We now present the proportion of employed patients and controls in Table 1.

6. Fig. 1 C legend: Lines 392-393 are not actually shown in the figure C. Need to either show the non-significant relationship in C, or adjust this statement in the legend.

[1] Cooper JA, Barch DM, Reddy LF, Horan WP, Green MF, Treadway MT. Effortful goal-directed behavior in schizophrenia: Computational subtypes and associations with cognition. J Abnorm Psychol 2019;128(7):710-722.

[2] Makary MM, Polosecki P, Cecchi GA, DeAraujo IE, Barron DS, Constable TR, Whang PG, Thomas DA, Mowafi H, Small DM, Geha P. Loss of nucleus accumbens low-frequency fluctuations is a signature of chronic pain. Proc Natl Acad Sci U S A 2020;117(18):10015-10023.

[3] Murray K, Lin Y, Makary MM, Whang PG, Geha P. Brain Structure and Function of Chronic Low Back Pain Patients on Long-Term Opioid Analgesic Treatment: A Preliminary Study. Mol Pain 2021;17:1744806921990938.

[4] Treadway MT, Bossaller NA, Shelton RC, Zald DH. Effort-based decision-making in major depressive disorder: a translational model of motivational anhedonia. J Abnorm Psychol 2012;121(3):553-558.

[5] Treadway MT, Buckholtz JW, Schwartzman AN, Lambert WE, Zald DH. Worth the 'EEfRT'? The effort expenditure for rewards task as an objective measure of motivation and anhedonia. PLoS One 2009;4(8):e6598.

---

## [Decision Letter · Decision Letter 1]

25 Jul 2025

Loss of Effort in Chronic Low-Back Pain Patients: Motivational Anhedonia in Chronic Pain

PONE-D-25-00836R1

Dear Dr. Geha,

We’re pleased to inform you that your manuscript has been judged scientifically suitable for publication and will be formally accepted for publication once it meets all outstanding technical requirements.

Kind regards,

Roya Khanmohammadi, Ph.D

Academic Editor

PLOS ONE

Additional Editor Comments (optional):

Reviewers' comments:

Reviewer's Responses to Questions

**Comments to the Author**

1. If the authors have adequately addressed your comments raised in a previous round of review and you feel that this manuscript is now acceptable for publication, you may indicate that here to bypass the “Comments to the Author” section, enter your conflict of interest statement in the “Confidential to Editor” section, and submit your "Accept" recommendation.

Reviewer #1: All comments have been addressed

2. Is the manuscript technically sound, and do the data support the conclusions?

Reviewer #1: Yes

3. Has the statistical analysis been performed appropriately and rigorously? 

Reviewer #1: Yes

4. Have the authors made all data underlying the findings in their manuscript fully available?

Reviewer #1: Yes

5. Is the manuscript presented in an intelligible fashion and written in standard English?

Reviewer #1: Yes

6. Review Comments to the Author

Reviewer #1: (No Response)

7. PLOS authors have the option to publish the peer review history of their article (what does this mean? ). If published, this will include your full peer review and any attached files.

**Do you want your identity to be public for this peer review?** For information about this choice, including consent withdrawal, please see our Privacy Policy .

Reviewer #1: No

---

## [Editor Report · Acceptance letter]

PONE-D-25-00836R1

PLOS ONE

Dear Dr. Geha,

I'm pleased to inform you that your manuscript has been deemed suitable for publication in PLOS ONE. Congratulations! Your manuscript is now being handed over to our production team.

Kind regards,

on behalf of

Dr. Roya Khanmohammadi

Academic Editor

PLOS ONE